# Use of Monoglycerides and Diglycerides to Mitigate Poultry Production Losses: A Review

**DOI:** 10.3390/vetsci11030101

**Published:** 2024-02-27

**Authors:** Stacie R. Appleton, Anne Ballou, Kevin L. Watkins

**Affiliations:** 1Alura, Inc., Durham, NC 27703, USA; 2Iluma Alliance, Durham, NC 27703, USA; 3FoodFirst LLC, Indianapolis, IN 46202, USA

**Keywords:** alternative antimicrobials, antiviral, monoglycerides, intestinal integrity, poultry, food safety, microbiome, anti-inflammatory, emulsifier, antibacterial

## Abstract

**Simple Summary:**

This review summarizes the available literature on the impact of monoglycerides and diglycerides on poultry productivity and health. Consumption of poultry meat and eggs continues to increase in all geographies. Monoglycerides and diglycerides are products that producers can use to help fill the demand for their products while meeting the needs of today’s consumers, retailers, and policymakers.

**Abstract:**

Consumer preference dictates not only what food is consumed but also how that food is produced. Ingredients in livestock feed that are not antibiotics, not genetically modified, and not of animal origin but that are nutrient-like tend to be more acceptable to consumers, retailers, and producers. Mono- and diglycerides (MDG) fit these criteria, are commonly used in food, and are generally recognized as safe. But beyond being emulsifiers and a source of energy, MDG are also known to have antimicrobial, immune, and microbiome modulation and angiogenic activity. MDG in broiler diets have been shown to impact a variety of immune-related functions such as the regulation of circulating antibodies, as well as decreased gene expression or protein concentration of pro-inflammatory cytokines like TNF-α, IL-1β, IL-6, and IFN-γ. In addition, MDG can affect metabolic function and intestinal integrity. Results of this review show that MDG can serve as antimicrobial and growth-supporting alternatives for reducing poultry production losses, improving resource utilization and meeting consumer demand for sustainably produced and safe animal protein.

## 1. Introduction

Global consumption of poultry meat is expected to increase by 17.5 percent between 2024 and 2032 [1]. The Food and Agriculture Organization of the United Nations (FAO) reported that terrestrial animal-source foods contribute to a healthy diet by providing energy, protein, fatty acids (FA), vitamins, and minerals that are less commonly available in non-animal-sourced foods [2]. While livestock play a critical role in improving food security, human health, and livelihoods, the FAO also notes that challenges such as high resource utilization, greenhouse gas emissions, food-borne illness, antimicrobial resistance, and animal welfare need to be addressed to ensure sustainability [2]. The global poultry industry continues to search for products and production methods that will help meet the demand for safe, affordable, and sustainably produced food while also reducing resource utilization. Today, ingredients in livestock feed that are not antibiotics, not genetically modified, and not of animal origin but that are more nutrient-like tend to be more acceptable to consumers, policymakers, and producers. Monoglycerides and diglycerides (MDG) have the potential to improve animal health, welfare, and productivity and reduce the prevalence of human food pathogens while lessening environmental impact and not contributing to antimicrobial resistance.

Antimicrobial lipids are comprised of a group of amphiphilic lipid molecules that have the capacity to directly impact bacteria, membrane-enveloped viruses, and some fungi by direct lysis of the cell membrane and a variety of additional mechanisms to disrupt cellular metabolism and function [3,4,5]. Some of the most studied lipids in livestock and poultry production applications are FA and glycerides. Fatty acids are a group of organic compounds comprised of varying hydrocarbon chains with a carboxylic acid group at one end [5]. Monoglycerides and diglycerides are composed of glycerol linked to one or two FA, respectively. Although MDG can be found in the lipid components of food and feed, concentrations in natural sources are usually low. Commercially available MDG are generally prepared through the interesterification of glycerol with edible triglycerides from vegetable oils [6]. The resulting products contain a mixture of predominately ⍺-monoglycerides, diglycerides, and free glycerol but may also contain smaller concentrations of FA and triglycerides. When used under good manufacturing practices, MDG are considered safe and are commonly added to human foods as emulsifiers, stabilizers, thickeners, texturizers, lubricants, and flavoring agents [6,7]. Monoglycerides are amphiphilic compounds, containing both hydrophilic (water-attracting) and lipophilic (oil-attracting) components, allowing them to interact with both water and oil molecules. This structure allows a reduction in surface tension between the immiscible substances, thus preventing separation and providing more stable emulsions. This is beneficial for many reasons from a nutritional aspect, two of the most common being the maintenance and integrity of the foodstuff to prevent oxidation as well as to make the nutrients, primarily energy, contained in the fat portion of the diet more available, especially for young animals with immature digestive systems. The emulsification power is referred to as the hydrophilic/lipophilic balance or HLB value. As the HLB value increases, the emulsifiers become more soluble in water and their function changes from being water-in-oil emulsifiers to being oil-in-water emulsifiers (Figure 1) [8]. Monoglycerides have an HLB value of 2.8–3.8 and are considered water-in-oil emulsifiers [9].

In addition to their value in human food production, MDG are known to have antimicrobial-, immune-, and microbiome-modulating, as well as angiogenic, activity making them useful ingredients for livestock and poultry production applications [10,11,12,13,14,15]. Commercially available MDG products used in livestock and poultry production are generally short- and medium-chained single fatty acid glycerides or proprietary mixtures of glycerides labeled as nutrients and/or emulsifiers. While FA are also seen as antibiotic and growth promoter alternatives for use in livestock and poultry production [16], this review focuses on mono- and diglycerides, which are considered to be more stable, have a lower odor, higher antimicrobial activity, and more bioactive form of fatty acid derivatives.

## 2. Nutrient Utilization, Metabolism, Growth, and Carcass Composition

The studies reviewed for this article used a variety of glyceride types and blends, varying feed inclusion rates and feeding durations, and different bird types and ages, which are probably, at least in part, responsible for the reported variability in response. The feeding of glycerides rarely resulted in any reported negative effects, so MDG appears to be safe even at inclusion levels greater than 1% of the diet. Regarding nutrient utilization, metabolism, growth, and carcass composition, the most consistent responses suggest that MDG increases protein and decreases fat deposition.

The addition of emulsifiers to poultry feed has been shown to improve fat utilization [17,18,19,20], particularly in young animals with immature digestive function and limited ability for micelle formation. Feeding monolaurate to broilers for 35 days increased the apparent ileal digestibility of leucine and arginine but did not affect the digestibility of other amino acids [10]. The body weight (BW), body weight gain (BWG), feed intake (FI), and feed conversion ratio (FCR) were not affected by treatment. Broilers fed 0.1% monolaurate had increased villus height in the duodenum and jejunum but treatment had no effect on ileal villus height. Jejunal crypt depth was decreased in broiler fed by 0.3% and 0.5% monolaurate but 0.1% monolaurate had no effect on jejunal crypt depth and there were no treatment effects on duodenal and ileal crypt depth. Serum HDL-cholesterol was increased in birds fed all levels of monolaurate and serum triglycerides were decreased at 0.1% and 0.3% inclusion levels. There were no treatment differences observed in serum total cholesterol, LDL-cholesterol, or alkaline phosphatase.

Broilers (Cobb 700) fed a mixture of monoglycerides (0.5% of diet) for 42 days had increased 0–14-day average daily feed intake (ADFI), reduced apparent ileal methionine digestibility at day 14, reduced breast meat marinade uptake, reduced breast meat marinade loss, and increased breast meat cooking loss but had no effect on BW, average daily gain (ADG), FCR, apparent ileal digestibility of all other 18 amino acids, dry matter digestibility, carcass or part yield, or any other measure of meat quality compared with broiler fed 0.25% formaldehyde [21]. Another study feeding a blend of medium-chain monoglycerides to yellow-feathered broilers for 70 days reduced the breast meat drip loss, increased the total superoxide dismutase activity, and improved the sensory panel taste scores for meat and meat broth, broth aroma, and meat tenderness although broth color received lower sensory scores [22]. Treatment did not affect the meat quality measures in terms of the pH, total antioxidant capacity, moisture content, total protein, intermuscular fat, or ash. A separate study in which a mixture of medium-chain monoglycerides were fed to yellow-feathered broilers for 70 days improved BW, FI, and livability but had no effect on FCR [12]. Serum insulin, triglycerides, and lipopolysaccharides tended to be decreased in birds fed glycerides while there were no treatment differences observed in serum leptin, peptide YY, glucose, and total cholesterol. The duodenal villus height-to-crypt depth ratio was increased in birds fed glycerides compared with controls.

A series of experiments found that a mixture of butyric mono-, di-, and triglycerides decreased FI during the starter period but had no effect on the overall 42-day FI, BW, FCR, and livability [23]. In a second experiment, broilers fed the same butyric glyceride blend had similar growth performance and livability compared with birds fed bacitracin methylene disalicylate (BMD). Bacitracin reduced the duodenal villi crypt depth compared with control birds, while birds fed 0.2% glyceride blend had increased crypt depth compared with BMD but similar crypt depth compared with non-treated controls. Glyceride-fed broilers had carcass weights and breast meat yields similar to birds fed BMD; however, birds fed the glyceride blend at the 0.2% rate had heavier carcass weights and breast meat yields compared with birds fed at the 0.1% rate. A third experiment showed that the same mixture of butyric glycerides tended to increase carcass weight but had no effect on breast meat yield compared with birds fed virginiamycin.

Antongiovanni et al. [24] also evaluated a mixture of mono-, di-, and triglycerides of butyric acid and reported that broilers fed 0.2% glycerides for the entire 35-day trial had increased BW gain compared with control birds. Broilers fed 0.35% or 0.5% glycerides throughout the study or fed 1.0% glycerides for the first 21 days had a BWG gain similar to that of the control birds. There were no treatment differences in FI but birds fed glycerides had improved FCR compared with controls. While carcass composition was not affected by treatment, birds fed 0.2% glycerides had shorter villi, longer microvilli, and increased crypt depth in the jejunum.

Two experiments were conducted to evaluate the effect of the dietary inclusion of a mixture of mono-, di-, and tributyrin on broiler growth performance, serum metabolites, and lipid catabolism gene expression. Glyceride treatment improved 0–28-day FCR but had no effect on overall (0–40 day) FI, BW, or FCR in Experiment 1 [25]. However, in Experiment 2, glyceride treatment resulted in increased BW and FCR in 20-day-old broilers. Also, in Experiment 2, total cholesterol, triglycerides, HDL cholesterol, fatty acid synthase, and acetyl-CoA carboxylase-β were reduced in glyceride-fed birds while the total serum protein increased. Similarly, glyceride treatment increased the total protein in jejunum tissue but not in liver or abdominal adipose tissue. Glyceride treatment decreased jejunum, liver, and adipose lipoprotein lipase as well as decreasing fatty acid synthase in liver and adipose tissue. Gene expression results suggested reduced jejunal synthesis, storage, transport, and secretion of lipids with increased hepatic oxidation of lipids in glyceride-treated birds. Consistent with glyceride-mediated changes in tissue metabolites and gene expression, birds fed glycerides had reduced abdominal fat weight compared with controls.

Bedford et al. [26] evaluated various mixtures, dietary inclusion rates (0.05%, 0.1%, 0.2%, and 0.3%), and phases of continuous and staggered feeding of monobutyrin or a monobutyrin- tributyrin mixture and generally found carcass composition and serum biochemistry results suggestive of a glyceride-dependent reduction in fat deposition and increase in muscle deposition. A mixture of 0.05% monobutyrin and 0.05% tributyrin fed continuously for five weeks demonstrated the greatest increase in breast muscle weight relative to body weight. No treatment differences were observed in BW, ADG, or FCR.

Broilers fed a commercially available blend of MDG (SILOhealth 104, SILO S.P.A., Florence, Italy) for 35 days had an overall ADG, BW, or FCR similar to control birds [11]. Breast muscle as a percent of BW was increased in birds fed 0.2% and 0.3% glycerides compared with controls but birds fed 0.05% and 0.1% glycerides had breast muscle yields similar to control birds. However, as the inclusion level of glycerides increased, there was a significant linear decrease in abdominal fat as a percentage of BW. Breast muscle gene expression of forkhead box protein O4 and myostatin, known inhibitors of protein synthesis, were reduced at all inclusion levels of glycerides compared with controls. In addition, gene expression of acetyl-CoA carboxylase-⍺, a fatty acid regulatory enzyme, tended to decrease and the expression of lipoprotein lipase, which catalyzes the hydrolysis of triglycerides, tended to increase in the abdominal fat of birds fed glycerides.

## 3. Antimicrobial Activity of Monoglycerides and Diglycerides

Each MDG has a unique antimicrobial impact associated with the fatty acid bound at the ⍺ and/or β positions. Understanding the predominant individual modes of action allows for increased efficacy and synergies to be created when the monoglycerides are blended for a broad-spectrum multi-faceted approach supporting poultry production.

The recent interest and investigation of antimicrobial lipids and more specifically MDG have resulted from the development of microbial resistance to antibiotics. As this is a global concern for human and animal health, it has been a prominent focus of antimicrobial lipid research with an indication that there is little to no development of resistance to antimicrobial lipids due in part to the variety of modes of actions [5,27,28] and synergies created by glyceride blends of various chain lengths. Davies and Davies [29] reported the successful growth of bacterial cultures with sublethal concentrations of antimicrobial lipids with no signs of resistance [29]. The array of structure–function modes of action coupled with the variety of interactions with the host gastrointestinal (GI) tissue and immune response suggest that long-term use of MDG will not lead to resistance or loss of activity. Monoglycerides have increased activity against common Gram-positive and Gram-negative poultry pathogens both in vitro and in vivo, as compared to their FA counterparts [30], which are quickly absorbed from the lumen of the small intestine [31]. The rate of dissociation in the gastrointestinal tract (GIT) is dependent on the pH level of the medium and the pKa of the individual acid [13]. Thus, to maintain efficacy in the distal areas of the intestine, they must be protected or encapsulated. Conversely, MDG are pH-independent due to the molecular structure and thus maintain integrity and function throughout the gastrointestinal tract as they are not dissociated [30].

Monoglycerides have a distinct advantage in dissociating bacterial cell membranes in that they form micelles more readily at lower concentrations than FA and, once in micelle formation, are more effective at permeabilizing the outer cell membrane of bacterial cells than FA [13]. A more unique antibacterial mode of action is the active uptake of monoglyceride through dynamic protein channels in the membrane called aquaglyceroporins. Once inside the bacterium, the FA dissociate from glycerol, releasing their cationic (H+) and anionic (A−) fractions. The anionic portion of the acid interacts directly with the bacteria’s nucleic acids, affecting the two-component regulatory systems, translation and transduction, and replication processes [32]. This impacts the pathogenic capacity (virulence) and the expression of resistance mechanisms in microorganisms. In addition, the cationic fraction (H+) of the acid decreases the internal pH of the bacteria, denaturing proteins and affecting the enzymatic activity, which decreases the metabolism of bacteria, leading to bacteriostasis. To counteract the drop in pH, the bacteria try to remove hydrogen ions through protein complexes and membrane enzymes (ATPases) that lead to critical energy expenditure, which in turn triggers the bacteria to actively import more glycerol as an energy source [13], thus initiating a negative physical, biochemical, and bactericidal cycle.

Individual monoglycerides have varying biological activity against pathogenic bacteria based largely on the physiochemical properties of the lipid in relation to the structure and function of biological macromolecules [5]. For example, glycerol monolaurate or monolaurin (GML) has been documented to have an equally bactericidal impact against Gram-positive, Gram-negative, and non-Gram staining bacteria [33]. However, organisms such as in the *Enterobacteriaceae* family, which have a specific lipopolysaccharide (LPS) composition, are more resistant to GML at neutral pH with their impact being improved as pH becomes more acidic [33]. Schlievert et al. indicated *E. coli* and *Salmonella minnesota* were two such organisms but that mutant *S. minnesota*, lacking the O side chain and other core components of the LPS layer, were extremely susceptible to GML [33,34]. Additional studies have confirmed that Gram-negative bacteria that have a lipooligosaccharide structure instead of the intact LPS are more sensitive to GML [5,33]. While the core of the LPS layer is relatively stable, the polysaccharide side chains are highly variable. The variation in the LPS structure also impacts the virulence of the strain or serotype [35]. Yoon et al. indicated distinct membrane disruptive impacts of anti-microbial lipids on phospholipid membranes due to individual physiochemical properties such as lipid charge [5]. Monoglycerides are nonionic and therefore impact the membrane translocation rate and membrane strain differently than anionic FA. Using the supported lipid bilayer (SLB) platform, the membrane morphological response to GML was the production of spherical bud formations while the response to lauric acid was to form elongated tubule structures. The antibacterial impact of these biophysical variations was documented with parallel minimum inhibitory concentration (MIC) studies of the impacts of GML and lauric acid on *Staphylococcus aureus*, with MIC being defined as the level at which no bacterial growth occurred [5]. GML is bactericidal against *Mycoplasma*, thus indicating that the peptidoglycan cell wall structure is not the primary mode of bactericidal action against this bacterium [33]. In addition, GML interaction with plasma membranes contributes to a variety of direct and indirect impacts on the two-component regulatory systems that bacteria use to interact with their environment. For example, GML has the ability to inhibit exotoxin production of Gram-positive bacteria by blocking signal transduction in a two-component regulatory system, thus interrupting transcription at sub-inhibitory levels [32,34,36]. This mechanism of multiple impacts along with the additional targets make bacterial resistance to GML improbable [33].

Some of the predominant antimicrobial modes of action for monocaprin have been reported as dissociation of the Gram-positive bacteria, *S. aureus* and *B. subtilis*, cell wall structure, and damage to the lipopolysaccharide layer of the cell wall of Gram-negative bacteria, *P. aeruginosa* and *E. coli*, under neutral or alkaline conditions with reduced antibacterial impact under slightly acidic conditions [37]. *Campylobacter jejuni*, *C. coli*, and *C. lari* were shown to be susceptible to monocaprin with greater susceptibility of *C. jejuni* at a lower pH of 4–5. The sensitivity of *Salmonella* spp. and *E. coli* to monocaprin was likewise shown to be enhanced by a pH of 4–5 [14]. Ma et al. demonstrated the *E. coli* ATCC 25922 membrane dissociation capability of monocaprin with SYOTX green fluorescent dye and reported the deterioration of the outer cell wall layer at sub-lethal doses [38]. Additional data supporting linear concentration-dependent outer membrane degradation of *E. coli* ATCC 25922 were reported by measuring the reaction of β-lactamase, an intracellular enzyme in the periplasmic space of Gram-negative bacteria with a colorimetric cephalosporin reagent, as well as the leakage of β-galactosidase, an enzyme reaction with O-Nitrophenyl-β-D-Galactopyranoside, which indicates the disruption of the cytoplasmic membrane [38]. Mitochondrial respiration provides an electrochemical proton gradient or protonmotive force that supplies the energy necessary for ATP production in the cell. Both the cytoplasmic proton gradient (charge) and pH gradient are required for function. The addition of monocaprin at twice the MIC to *E. coli* isolates indicated immediate cytoplasmic membrane depolarization and collapse of the pH gradient [38]. These results agree with additional reports that monocaprin was shown to be active against the lipid bilayers of the LPS platform and increased bilayer fluidity [5]. An additional point of interest reported by Yoon et al. was that varying forms of monocaprin induced different membrane morphological responses. Specifically, monomeric monocaprin produced elongated tubules while micellular monocaprin produced spherical buds [39].

Butyric acid and the esterified form, monobutyrin, are some of the most well-studied SCFA. Monobutyrin has many in vivo functions that directly and indirectly promote animal health in addition to its antimicrobial functions [40]. In addition to its antimicrobial effects, monobutyrin has a direct impact on gut health as it can be utilized directly by intestinal epithelial cells [41] and it reduces enteric inflammation via reduction in expression of TNF-α [42]. Butyrate glycerides were shown to mitigate the negative impact of lipopolysaccharide (LPS) challenge on histomorphometry of the ileum and support intestinal immune response [43]. In addition, monobutyrin provides improvement in barrier function and support of tight junction proteins after challenge [44].

The unique properties and modes of action of antimicrobial lipids are very similar in some cases and even overlap but are not identical to others [27]. As such, Gram-positive and Gram-negative bacteria have varying susceptibility to different individual antimicrobial lipids and blends. For example, Gram-negative bacteria are affected by SCFA and its derivatives, generally C6 or less [45], and can also be impacted by mixtures of MCFA and monoglycerides, such as caproic and caprylic acids [28], but are not affected by long-chain unsaturated fatty acids [46]. Synergistic and additive impacts have been suggested since the early study of antimicrobial lipids, between varying combinations of FA and their derivatives as well as between the antimicrobial lipids and the gastrointestinal system during digestion, but need additional work to be completely understood [47,48].

In contrast to many pathogens, commensal lactic acid-producing bacteria species such as *Lactobacillus*, *Bifidobacterium*, and *Bacteroides* are generally more acid-tolerant [3,49]. As will be discussed in more detail later in this review, MDG support the commensal populations, which also use glycerol as an energy source, by improving population consistency and diversity. In addition, modulation of the microbiome diversity by the inclusion of monoglycerides offers immune support and improved anti-inflammatory and anti-oxidative capabilities that mitigate intestinal insult and promote animal health and resistance to challenge [3].

## 4. Antiviral Activity

Similar to their antibacterial activity and specificity, glycerides, as membrane disrupters, have been shown to inactivate enveloped viruses. Welsh et al. identified lipid fractions and monoglycerides as a source of antiviral activity in human milk [50,51]. Thormar et al. compared the antiviral activity of FA and monoglycerides against three enveloped viruses (vesicular stomatitis, herpes simplex, and visna) and found that medium-chain saturated and long-chain unsaturated FA had significant antiviral activity while short-chained and long-chained saturated FA had no or very low activity [52,53]. However, monoglycerides had the most antiviral activity and showed activity at concentrations 10 times lower than that of the corresponding FA.

More recently, it was reported that monolaurate added to feed (0.25% to 2.0% inclusion rate) was a more potent inhibitor of African swine fever virus infectivity than an FA blend [54]. Phillips et al. reported that while a blend of MCFA (1.1%), a mixture of formaldehyde and propionic acid (0.342%), and a blend of monoglycerides (0.165, 0.276, and 0.386%) all demonstrated about a 2-log reduction in virus recovery from feed inoculated with porcine epidemic diarrhea virus, the monoglyceride blend appeared effective at lower concentrations [55]. When virus-inoculated feed was fed to pigs for 20 days, 58% of control pigs tested positive for the virus while no pigs receiving feed with added MCFA or monoglycerides tested positive. Some have concluded that monoglycerides should be seen as an important class of feed additives for the protection of pig productivity and health and feed pathogen mitigation [55,56].

While few studies were found regarding the antiviral activity of glycerides in poultry, caprylic acid, sodium caprylate, and monocaprylin were shown to inactivate avian influenza virus in chick embryonic kidney cells [57]. As reported by others, monoglycerides demonstrated higher antiviral activity than the corresponding free FA or FA salt. While monocaprylin reduced avian influenza viral particles 2 to 3 log at 0.08% and 0.16%, respectively, it took 0.2% to 0.4% caprylic acid for the same level of inactivation. Sodium caprylate did not demonstrate significant antiviral activity. While the previously reviewed research shows that glycerides can have direct antiviral activity, Mellouk et al. reported that lauric acid glycerides added at 0.33% of feed can enhance broiler’s resistance to infectious bronchitis through the mechanism of an improved humoral immune response [58].

## 5. Effects on Intestinal Morphology, Gut Immunity, and the Microbiome

The impact of the various MDG on GI morphology, gene expression, and microbiome composition are inextricably tied to the growth, carcass composition, and performance benefits that some studies have found. However, the mechanisms of action of MDG in this regard are poorly described, particularly in poultry. While micellar aggregation and subsequent disruption of microbial membranes are certainly important in the control of bacterial and viral pathogens, the role of this mechanism in broader alterations of the microbiome is less explored [5,39]. Several studies in the past 10 years have shown that MDG consistently increases levels of SCFA-producing bacterial species, sometimes accompanied by additional changes in composition and diversity [12,59,60,61,62,63]. With regard to the impact on the host animal, MDG and FA have repeatedly been shown to stimulate the production of host genes and proteins related to lipid metabolism, oxidation, and immune function [64,65].

### 5.1. Bacterial Patterns with MDG

As interest in the beneficial effects of MDG has grown over the years, much of that interest has focused on their potential to provide nonantibiotic control of gut pathogens [13]. However, trials both with and without pathogen challenges often demonstrate similar changes in commensal bacterial populations, suggesting that MDG can modulate the microbiome through unknown mechanisms that are not dependent on the disruption of pathogenic populations. Liu et al. reported increased levels of SCFA-producing bacteria when feeding a mix of monolaurate and monocaprylin to laying hens, specifically *Romboutsia* and *Lachnospiraceae* spp. They also reported a decrease in *Proteobacteria* [61]. Further studies from the group suggest that these commensals are consistently impacted by various types of MDG, as well as other groups such as *Bifidobacteriaceae* and *Bacteroides* [12,60]. In addition to changes in SCFA-producer abundance, levels of acetic, propionic, butyric, isobutyric, and isovaleric acids were also increased, supporting the role of these microbes in the production of beneficial metabolites. Studies from other research groups have corroborated the general increase in SCFA producers following supplementation with MDG [63,65].

Beyond increasing the abundance of specific beneficial bacteria, several studies have noted another desirable outcome of MDG supplementation: increased community diversity. Several metrics of community diversity (⍺-diversity) exist, each evaluating slightly different facets of what it means to have a rich and diverse population. While the existing literature does not argue that high ⍺-diversity is beneficial per se, there is a growing body of evidence associating a variety of disease phenotypes with abnormally low ⍺-diversity [66,67,68,69]. Increased community richness and phylogenetic diversity may be indicators of balanced communities with redundancy in metabolic and structural function. This balance could make the community more resistant to outside perturbations or allow for faster recovery when its environment does change. Kong et al. tested several dose levels of glycerol monolaurate (0.03–0.12%) in broilers for 14 days [63]. In addition to increasing the abundance of several beneficial bacteria, the different levels of MDG increased microbial diversity in a dose-dependent manner in the first week of life. Another study in mice, also evaluating various levels of monocaprylate, demonstrated a clear association between MDG and several ⍺-diversity metrics [65].

Not all support for the role of MDG in microbiome modulation comes from poultry. Much of the work being performed to explore the mechanisms of action of MDG is in other animal models. Many of these studies suggest that the impact of MDG is similar in monogastric mammals. Zhang et al. reported increases in *Bifidobacterium pseudolongum* in mice following supplementation with glycerol monolaurate at 0.16%. Zhao et al. tested monocaprylate (0.015 versus 0.16%) in mice as a food preservative. They reported increased *Lachnospiraceae*, *Lactobacillus*, *Ruminococcus,* and *Clostridiales*, depending on dose, with increased microbial SCFA levels. As mentioned earlier, they also reported increased ⍺-diversity [59,62].

It is important to note that drawing conclusions from these studies is made more difficult by the fact that there is a wide range of blends and concentrations represented in these studies. Not only that, while some studies feature health challenges such as necrotic enteritis, others do not. Studies in which a health challenge is a part of the experimental design seem more likely to demonstrate differences in host parameters, i.e., immune activity or performance outcomes [65,67].

### 5.2. Gut Integrity and Function with MDG

Understanding the impact of these products on the microbiome is only part of the story. Even more important is the direct and indirect impact of MDG on the GI itself. As a major site of immune activity and interaction with microbiota, the gut plays a critical role in determining the animal’s response to pathogen challenges, both in and out of the gut. Furthermore, there is mounting evidence that MDG have metabolic impacts outside the incidental energy they may add to the diet. Studies in broilers and laying hens have shown that supplementation with varying levels of MDG can stimulate changes in levels of hormones related to reproduction and satiety, as well as a host of genes related to tight junctions, lipid metabolism, and immune function [61,64,65].

Relatively few studies have evaluated MDG in laying hens but one study demonstrated improved performance parameters (egg weight and mass and feed conversion) that could be attributable to increased serum levels of reproductive hormones like luteinizing hormone, estradiol, and follicle-stimulating hormone, as well as increases in circulating triglycerides and glucose [61]. Other studies in broilers, some featuring GI pathogen challenges, others unchallenged, demonstrated consistent increases in genes related to tight junctions, decreased inflammation, and increased transport of nutrients. Sacakli et al. fed MDG to broilers at either 0.03% or 0.05% and showed transitory improvements in BWG but consistent increases in zona occludens 1 (ZO1), sodium-glucose co-transporter 1, and peptide transporter 1. While they measured no differences in histomorphometry, they did observe an increase in the number of goblet cells in the jejunum, denoting the capacity for higher levels of mucus production [64]. In two studies of MDG in broilers featuring a challenge (necrotic enteritis model, LPS challenge), the authors highlighted increases in serum antibodies as a primary outcome of the MDG [65,70]. Furthermore, Kong et al. observed that MDG (monolaurate at 0.12%) restored the expression of ZO1 and occludin to unchallenged levels [65].

### 5.3. Gut Immunity

In addition to impacting gut function and metabolism, MDG have regularly been shown to promote an anti-inflammatory profile, particularly in the presence of an immune challenge. A butyrate-heavy MDG mix in broilers demonstrated dose-response in expression of IL1B and TNFa genes and, as reported above, also impacted measurements related to metabolic function and integrity in the gut (SGLT1, PePT1, and goblet cell numbers) [64]. Other studies have reported increased antibodies in circulation, sometimes associated with decreased gene expression or protein concentration of pro-inflammatory cytokines like IL-1β, IL-6, and IFN-γ [63,65,70].

In mice, Zhang et al. demonstrated that monocaprylate stimulates IL10 and suppresses IL1B in the circulation, as well as impacting the expression of other genes such as PPAR-γ, that can have broad-reaching effects on immune function [59]. Also, monolaurate treatment in mice fed a high-fat diet reduced circulating pro-inflammatory cytokines typically associated with high-fat diets (TNF-ɑ, IL-6) and reduced expression of MyD88 and TNF-ɑ genes [62].

### 5.4. Free Fatty Acid Receptors (FFAR)

Though none of the primary research studies cited in this paper measured FFAR, this class of G-protein-coupled receptors could be playing a major role in mediating the host response to MDG. Though often thought of as taste receptors, these receptors are expressed on a large variety of cells not associated with taste sensing but rather, nutrient sensing. Officially, FFAR2 appears to be the only reported FFAR gene in the chicken genome, while many mammal genomes have four identified FFAR genes [71,72]. However, some studies have evaluated the expression of other putative FFAR genes in chickens and found them to be associated with oral and intestinal nutrient sensing; serotonin expression and behavior; PPAR-related gluconeogenesis; and immune cell regulation [73,74,75,76]. In mammals, the different FFAR genes have an affinity for FA of different lengths. While little work has been conducted in chickens to determine the specific affinity of the known and putative FFAR genes, this could represent a mechanism by which FA from MDG in the avian gut are stimulating systemic and local changes in hormone production, immune activation or suppression, and GI tight junction integrity [77].

## 6. Practical Applications

As researchers continue to investigate modes of action and functionality, producers must consider the practical application and potential benefits of MDG. Glycerides are effective emulsifiers that can help to increase nutrient availability, especially in young birds, and can improve enteric health to further enhance nutrient utilization and productivity. Antimicrobial activity along with immune and microbiome modulatory effects of MDG have the potential to enhance pathogen control throughout the production cycle leading to improved animal health and food safety.

Since the reported bioactivity of MDG provides producers with a variety of approaches to mitigate poultry production losses, having clear objectives or treatment targets will help with the selection of the product, dose, and use program. Producers with clear goals such as mitigation of a specific health condition, reducing pre-harvest pathogen load, or improving gut health during periods of peak challenge will design more targeted programs with better opportunities for success. As discussed in this review, individual MDG have unique applications, functions, and dose responses [56] and blends can create synergistic effects [13]. As such, selecting the correct glyceride or glyceride blend and using the proper dosage during the appropriate production phase is necessary to achieve the best return on your MDG investment.

## 7. Summary

While MDG are shown to improve growth performance, nutrient utilization, carcass characteristics, enteric health, response to disease, and food safety in poultry, literature reports are inconsistent. Some of these inconsistencies are most likely related to differences in the type of glyceride or glyceride blend, dietary inclusion rate, treatment timing, and testing protocols (e.g., disease challenge or not) as indicated in Table 1. However, while efficacy tended to be glyceride-, dose- and feeding program-dependent, there were some commonalities in the literature that can provide producers with practical guidance on how to use or test MDG in their operation.Most in vivo responses are observed when the diet contained between 0.05% and 0.5% MDG and inclusion rates as high as 1% showed no adverse effects;Nutrient utilization, carcass composition, and serum biochemistry results tend to point toward glyceride-dependent reductions in fat deposition with concurrent increases in muscle deposition;Antimicrobial activity is seen against many of the problematic organisms found in poultry production and meat processing including *C. perfringens*, *E. coli*, *Salmonella* spp., *Campylobacter* spp., *Mycoplasma* spp., and some enveloped viruses as illustrated in Table 2;There tends to be an increase in the abundance of specific beneficial bacteria (i.e., SCFA producers) and a general increase in intestinal community diversity;Increases in overall enteric health and function were commonly reported through a variety of mechanisms including improved tight junction integrity, increased villus height, and reduced inflammation.

While this review presents several examples of how MDG can provide a nutraceutical-type approach to reduce poultry production losses, there are still some knowledge gaps related to the ideal glycerides, glyceride mixes, dosages, and feeding phases that can provide the best efficacy and cost–benefit ratio to the producer. There was also a lack of data related to the use of MDG in the water, which, under some conditions, may provide a more targeted and practical treatment opportunity than feed supplementation.

## Figures and Tables

**Figure 1 vetsci-11-00101-f001:**
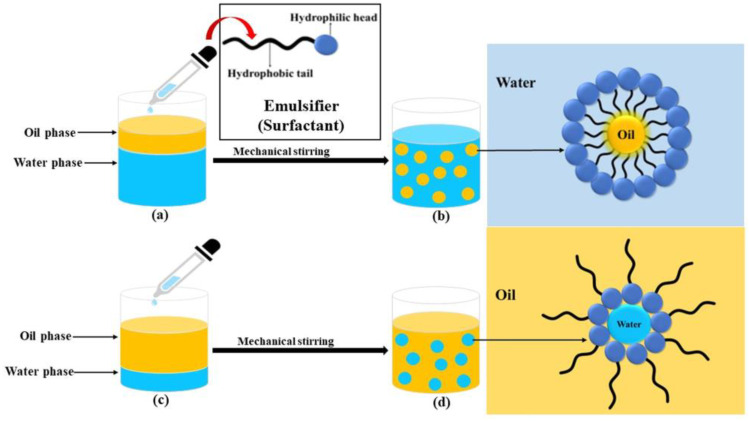
Representation of oil in water and water in oil emulsions. Courtesy of Singh and Pulikkal, 2022 [8]. (**a**) Oil in water before addition of emulsifier. (**b**) Formation of oil in water emulsion. (**c**) Water in oil before addition of emulsifier. (**d**) Formation of water in oil emulsion.

**Table 1 vetsci-11-00101-t001:** Summary of inclusion rates for various in vivo monoglyceride applications cited in the review.

Product Description	Animal Age, d	Inclusion Rates, %	Response Criteria in Broilers Except Where Noted	Literature Reference
Butyric, caprylic and capric acids	0–42	0.03, 0.05	gut structure and growth performance	[64]
Butyrate monoglycerides	0–28	0.015	intestinal barrier function and inflammation responses in mice	[43]
Butyrate mono-, di- and tri-glycerides	0–42	0.2, 0.4	growth performance, and carcass yield	[23]
		0.1, 0.2		
	0–27	0.4		
	0–42	0.2, 0.35, 0.5, 1.0	growth performance, intestinal morphology, and carcass yield	[24]
	0–7	0.6	growth performance, carcass composition, serum biochemistry, and gene expression	[25]
	8–20	0.3		
Glycerol monocaprylate	28–154	0.015, 0.16	gut microbiota, short-chain fatty acid production, metabolism, and inflammation in mice	[59]
Glycerol monolaurate	0–112	0.16	gut microbiota and regulating glycerophospholipid metabolism in mice	[62]
	0–14	0.03, 0.06, 0.09, 0.12	growth performance, immunity, intestinal barrier, and cecal microbiota	[63]
	0–21	0.12	immunological stress and intestinal mucosal injury	[65]
Mixture of butyric, propionic, caprylic, capric, and lauric acid mono- and diglycerides	0–35	0.05, 0.1, 0.2, 0.3	growth performance, carcass composition, fat, and muscle gene expression	[11]
Mixture of medium chain alpha-monoglycerides	0–42	0.03, 0.045, 0.06	community and function of gut microbiota	[60]
	280–448	0.03	productive performance, egg quality, serum biochemical indices, and gut microbiota in laying hens	[61]
Mixture of monoglycerides	1–14	0.5	growth performance, nutrient utilization, and meat yield	[21]
	14–28	0.4		
	28–42	0.2		
	42–49	0.15, 0.25, 0.35	viral disease transmission to piglets	[56]
Mixture of mono- and di-glycerides	0–43	0.5	growth performance, nutrient digestibility, and energy utilization	[17]
Monobutyrin	0–28	0.25, 0.75, 1.25	lipid metabolism and gut barrier function	[44]
Monobutyrin and/or tributyrin	1–35	0.05 tributyrin + 0.05 monobutyrin	growth performance, carcass composition, and blood biochemistry	[26]
		0.05 tributyrin + 0.2 monobutyrin		
	1–7	0.05 tributyrin		
	8–14	0.05 or 0.2 monobutyrin		
	15–35	no glycerides		
Monolaurate	3–35	0.1, 0.3, 0.5	growth performance, nutrient utilization, and blood biochemistry	[10]
Monolaurate and Monodecanoate	0–70	0.03	growth performance, intestinal microbiota, intestinal morphology, and blood chemistry	[12]
			carcass composition	[22]
Monostearate	18–42	0.02	growth performance, carcass composition, serum biochemistry, and liver biochemistry	[20]
Ricinoleate	0–21	0.035, 0.07	growth performance, nutrient metabolism, and blood chemistry	[18]
	22–39	0.028, 0.056		

**Table 2 vetsci-11-00101-t002:** Summary of concentrations for various in vitro monoglyceride applications cited in the review.

Product Description	Inclusion Rates, %	Response Criteria	Literature Reference
Butyrate glycerides	0.025, 0.7	antibacterial activity against *Salmonella typhimurium* and *Clostridium perfringens*	[40]
Monobutyrin	20 pg	angiogenesis in chick chorioallantoic membrane	[15]
Mixture of monoglycerides	0.05–2	antibacterial activity against *E. coli* and *Enterococcus*	[48]
Monocaprylin	0.2, 0.4	anti-viral activity	[55]
Monocaprin, monolaurin, monomyristin, monopalmitin, and monostearin	0.016–1.0	antimicrobial activity	[37]
Monocaprylate, monocaprin, monolaurin, and monomyristin	0.5	antimicrobial activity	[28]
Monolaurate	2	antimicrobial activity	[32]
	0.05–0.1	antimicrobial activity	[33]
	0.1–0.3	antimicrobial activity	[34]
	0.01	antimicrobial activity	[36]

## Data Availability

No new data were created or analyzed during the development of this manuscript.

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
