# Peer review of "Use of Monoglycerides and Diglycerides to Mitigate Poultry Production Losses: A Review"

_vetsci, 2024, doi:10.3390/vetsci11030101_

Round 1
Reviewer 1 Report
Comments and Suggestions for Authors
This is a difficult topic to review and the authors have done a good job of doing this, so well done!
Comments on the Quality of English LanguageI have a few comments on the English largely around the use of the comma prior to the word and. It is grammatically correct to use a comma before "and" (and other coordinating conjunctions such as "but", "or", "nor") only when it splits two independent clauses (i.e. if you remove the "and" you will be left with two complete sentences), or if you're using it as an Oxford comma. The use of the Oxford comma is a matter of style, meaning that some publishing styles stipulate its use while others don’t. In other words, it’s not incorrect to use the Oxford comma or not to use it, but it is advisable to be consistent one way or the other. AP style—based on The Associated Press Stylebook, the style guide that American news organizations generally adhere to—does not use the Oxford comma. I'm not sure what MDPI standard is but, for me I would remove it if it isn't needed, starting with line 17 in the Abstract.
Line 72: Remove "the" from the sentence.
Line 98 and 99: Remove the verb "marinate" and replace with the noun form "marinade".
Author Response
Thank you for the kind words regarding the manuscript and coverage of the topic, these are greatly appreciated. Regarding the edits, the oxford commas have been removed, the removal of the word “the” on line 72 and the change of the word marinate to marinade was completed. We greatly appreciate your time and input. Please feel free to reach out if you have questions or need additional information.
Reviewer 2 Report
Comments and Suggestions for Authors
This review observes the current knowledge of the use of mono- and diglycerides in the poultry industry. The review is quite broad and covers many areas of the possible effects of the use of these compounds in poultry production. On the whole, the manuscript is well written, scientifically sound, and well organized. The only comment/suggestion the reviewer has is that the inclusion of an explanatory Table listing the uses and effects of these compounds on the birds themselves and in the poultry industry would be quite helpful as a quick reference and guide both while reading the manuscript and when information on these compounds needs to be quick- referenced.
Author Response
Thank you for the kind words regarding the preparation of the manuscript and the broad application of the information presented for the industry. In response to the request for a summary table, we have created and added that to serve as a quick reference for readers. We greatly appreciate your time and input. Please feel free to reach out if you have questions or need additional information.
Reviewer 3 Report
Comments and Suggestions for Authors
There had been no recent reviews on this subject and this is a worthwhile contribution to the field of poultry nutrition and health. It is well written and provides a good summary of the state of art.
Three comments, which needs to be considered in any revision.
Section 2: The basis behind the emulsifying properties need be discussed.
A major weakness is the lack of tables/ figures, which would have been instructive and strengthen the discussion. Suggest inclusion of tables/figures to emphasise the key points in each section/ sub-section.
Authors repeatedly state that ‘inclusion rates as high as 1% showed no adverse effects’. This is redundant. These components are natural and there should be no adverse effect even at 10% inclusion.
Author Response
- Thank you for the kind words regarding the manuscript topic and benefit for the industry, as well as for the manuscript preparation. Regarding the suggestions for the inclusion of tables and figures, we have included a summary table as well as an explanatory diagram to help support the other suggested revision of a more in-depth explanation of emulsifiers. Regarding the comments on the inclusion rates, we understand your perspective and agree. No major changes to the manuscript were made, however, as we were citing existing literature. Please advise if you have further suggestions on this. We greatly appreciate your time and input. Please feel free to reach out if you have questions or need additional information.
Round 2
Reviewer 3 Report
Comments and Suggestions for Authors
Thanks for careful changes and the revision.